# Energy Efficiency Model Construction of Building Carbon Neutrality Design

**Rui Liang** [1]**, Xichuan Zheng** [1] **, Jia Liang** [2] **and Linhui Hu** [3],*

1   School of Architecture and Urban Planning, Guangdong University of Technology, Guangzhou 510062, China; liangrui@gdut.edu.cn (R.L.); 2112110009@mail2.gdut.edu.cn (X.Z.)
2   School of Management and Economics, The Chinese University of Hong Kong, Shenzhen 518000, China; 120090748@link.cuhk.edu.cn
3   School of Art and Design, Guangdong University of Technology, Guangzhou 510062, China
*   Correspondence: hlh@gdut.edu.cn

**Abstract:** We aim to create a feasible quantitative method to calculate the energy efficiency of building designs that are carbon-neutral and to develop a workable way of calculating energy efficiency in buildings that achieve carbon neutrality and the system for such a building's design energy efficiency function. This paper first clarifies the idea of the design energy efficiency function for a carbon-neutral building over its whole life cycle. Subsequently, through the efficient analysis of carbon-neutral design dimension measures, this paper summarizes and integrates the mature theories of various disciplines, puts forward the energy efficiency function model of carbon-neutral design background, propulsion, and coverage, and implements the energy efficiency function model of carbon-neutral design in the whole life cycle of buildings. The index value of a building's carbon emission factor is established based on the carbon accounting factor published by the Intergovernmental Panel on Climate Change, and a carbon neutrality energy efficiency model for buildings over the duration of their whole life cycle is constructed. The results were as follows. 1. Technology energy efficiency is far better than scale energy efficiency and comprehensive energy efficiency. 2. The better the energy efficiency value inside the building stage, the less consumption and the higher the production. 3. Construction is when technical energy is used the least. This paper refers to a systematic design method that makes the level of building carbon neutrality design technologically advanced with the aid of all types of big data related to the building life cycle and various innovative design theories in order to fully represent the fundamental level, development potential, and the effectiveness of choosing the strategy of building carbon neutrality.

**Keywords:** full life cycle; building carbon neutrality design; energy efficiency function; model

## 1. Introduction

The carbon footprint of a building must be determined since it has a substantial impact on both energy use and carbon emissions [1]. The quantity of external energy needed while a structure is in use is referred to as "building energy consumption" in accordance with the Energy Consumption Standard for Civil Buildings (GB/T51161-2016) [2]. By 2022, over 50% of China's total carbon emissions will be attributable to all carbon emissions related to construction. The built environment is responsible for almost half of the global annual carbon dioxide emissions. The annual contribution of building materials and structures to these total emissions is 20%, whereas the annual contribution of construction operations is 27% [3,4]. Sustainable development and low-carbon building techniques are now widely accepted ideas. It basically takes a long-term plan that must be effective in social, environmental, and economic aspects to create a clean, low-carbon, and sustainable environment in public buildings [5]. However, while having a high initial cost, it provides long-term benefits in terms of building performance and low-cost maintenance, which benefit both the environment and building occupants [6]. In order to address the effects

of carbon neutrality on the construction sector, the British Construction Steel Association (BCSA) and Tata Steel (TS) have investigated and determined the best cost-effective mix of materials and technology to realize low-carbon and zero-carbon structures. The paper's findings suggest that whether a structure is composed of steel, concrete, or even wood, its construction type has little bearing on its carbon emissions [7]. Through building standards, which include fuel and electricity conservation, the British government directs buildings to produce zero carbon emissions [8]. The research project's cost analysis reveals that after it surpasses 40%, the building costs associated with each percentage increase are highly expensive, although the mix of various construction technologies is different. There are numerous ways to accomplish this, some of which are more affordable than others. The decision-making process in the past greatly affects the potential route, capital expense, and life cycle of achieving zero carbon output. Li (2021) developed a life cycle inventory-based carbon emission model and performed a statistical analysis of the carbon emissions of various building types [9]. The findings indicate that the envelope has a significant impact on a building's carbon emissions, but no research has been done on the impact of other influencing indicators. Residential building numbers in rich and emerging nations were estimated by Wang et al. (2021) [10]. The findings indicate that residential buildings have a lower impact in underdeveloped nations than they do in industrialized nations. The geographic location, level of technical expertise, and level of economic development are the influencing indices of architectural evaluation. A statistical examination of the energy use of 73 buildings throughout the course of their whole lives was conducted by Yan et al. in 2022 [11]. The findings demonstrate that the building use stage comprises 85% of the entire life cycle and has the highest energy use.

The summary of research conducted in China and other nations demonstrates that there are still numerous issues with the study of buildings that are carbon-neutral and energy efficient. Firstly, the research findings of various scholars are highly dissimilar because of the division of the life cycle and the carbon neutrality energy efficiency model. Carbon neutrality design techniques and carbon neutrality energy efficiency function models are two rare examples of carbon neutrality energy efficiency models. To further unlock the potential of developing carbon neutrality, it is required to establish a highly practicable calculation method for building carbon neutrality and energy efficiency. The innovation of this article is that the mathematical model of "energy efficiency" is the concept, theory, method, and tool of the innovation system of building a carbon-neutral design, combined with the actual development of the national carbon-neutral building target and studies the technical support measures of building design. Based on this, an energy efficiency model can be implemented to quantitatively predict the carbon-neutral design scheme of buildings. The "function" model quantitatively studies the operating rules and principles of building carbon-neutral energy efficiency, that is, taking the theoretical modeling of building carbon-neutral design evaluation as the research object by combining architecture, engineering, mathematics, and other disciplines, using the functional model of carbon neutrality design foundation, promotion, and coverage, and combining with the mechanism of carbon neutrality design concepts, the "function" model quantitatively studies the operating rules and principles of building carbon neutrality design efficiency. This is done by taking the theoretical modeling of building carbon neutrality design evaluation as the research object.

## 2. Theoretical Basis and Method Research

### 2.1. Concept of Energy Efficiency Function for Carbon Neutrality Design of Whole Life Cycle Buildings

The phrase "full life cycle of a building" refers to the entire procedure of producing materials and components, planning and designing, building and moving them, operating and maintaining them, and removing and treating them (waste, recovery, reuse, etc.). Significant technology content, a protracted construction period, high risk, and the involvement of numerous units are hallmarks of construction projects [12]. As a result,

this study utilizes the whole life cycle hypothesis to determine the carbon emissions of buildings throughout the course of their whole life cycle. The period from the structure's acceptance at completion to the end of its intended service life is referred to as the "whole life cycle of a building". The entire service life of typical constructions is longer than the anticipated service conditions. Additionally, the methods of architectural design and the key influencing factors at different stages of its entire lifecycle affect the maintainability of building structures, resulting in some structures having lower actual service lives than expected [13]. A specific definition and a general notion are both used to describe the concept of building energy use. The energy used by all materials and equipment in the building during manufacture and processing, as well as the operating energy used by the building while it is being used, are all included in the generalized building energy consumption. Building energy consumption, when interpreted strictly, only refers to the amount of energy used by a building while it is occupied, ignoring production energy consumption. Therefore, the term "full life cycle building energy consumption" refers to the whole operating amount of energy consumption from completion acceptance to design service life [14].

Building material preparation, construction, building use and maintenance, and building demolition were the four stages that Wolf et al. (2021) identified in the building life cycle [15]. Building energy consumption is broken down into three categories by Pauliuk et al. (2021): physical energy, chemical energy, and operational energy, or the three stages of materialization, usage, and demolition [16]. Construction projects, operations, maintenance, building deconstruction, and solid sewage disposal were the five stages that Ferreira et al. (2020) divided the building life cycle into [17]. By accounting for the manufacture, transportation, construction, operation, maintenance, and demolition of building materials, Hossain et al. (2022) divided the carbon emission during the full life cycle into six stages [18]. The four stages of a building's life cycle, which are manufacturing of building materials, construction, usage and maintenance of the building, and demolition, are used in this study's calculations. The energy consumption of buildings during use accounts for 50% to 70% of the total energy consumption. The calculation boundary for each stage is precisely established, and as a result, the carbon footprint of the building's life cycle is assessed. The design energy efficiency index of carbon neutrality for the entire life cycle of a building is shown in Figure 1.

Figure 1 illustrates the carbon dioxide produced during the production of building components as a result of the use or decomposition of raw materials and energy consumption. This evaluation of the energy efficiency of carbon-neutral design across the entire life cycle of structures is done to determine how efficient the design is. Carbon dioxide is released during building construction as a result of mechanical operation and energy consumption from various construction tasks; Carbon dioxide is released as a result of energy consumption during building operation and maintenance; The energy used for building demolition, filling compaction, and garbage recycling results in the emission of carbon dioxide.

The main method of achieving carbon neutrality in buildings is to offset operating carbon emissions through energy efficiency, emission reduction, and negative carbon technology [19]. In order to achieve zero carbon emission across the whole life cycle of a structure, the second technique involves analyzing the carbon footprint and offsetting ways from the construction, installation, and operation of building supplies through the final demolition and recycling. The key to achieving these three levels is the digital quantification of building carbon emissions in order to achieve targeted emission reductions. Table 1 illustrates the design procedure for a structure that is carbon-neutral.

To promote the carbon neutrality of buildings, the actions that must be taken in all areas of the construction industry are listed in Table 1. For instance, low-carbon building materials are used in the production stage after traditional building materials have been decarbonized. Encourage the usage of assembled construction and digital technology during the construction phase. Energy substitution, electrification, and energy efficiency

improvements should be implemented as much as possible during operation; during demolition, work should be done to optimize the demolition plan and recycle as many building components as possible. On the other hand, the foundation of achieving building carbon neutrality is to finish carbon quantification with reasonable accuracy, choose the best course of action, and guarantee the fairness and traceability of emission reduction data [20].

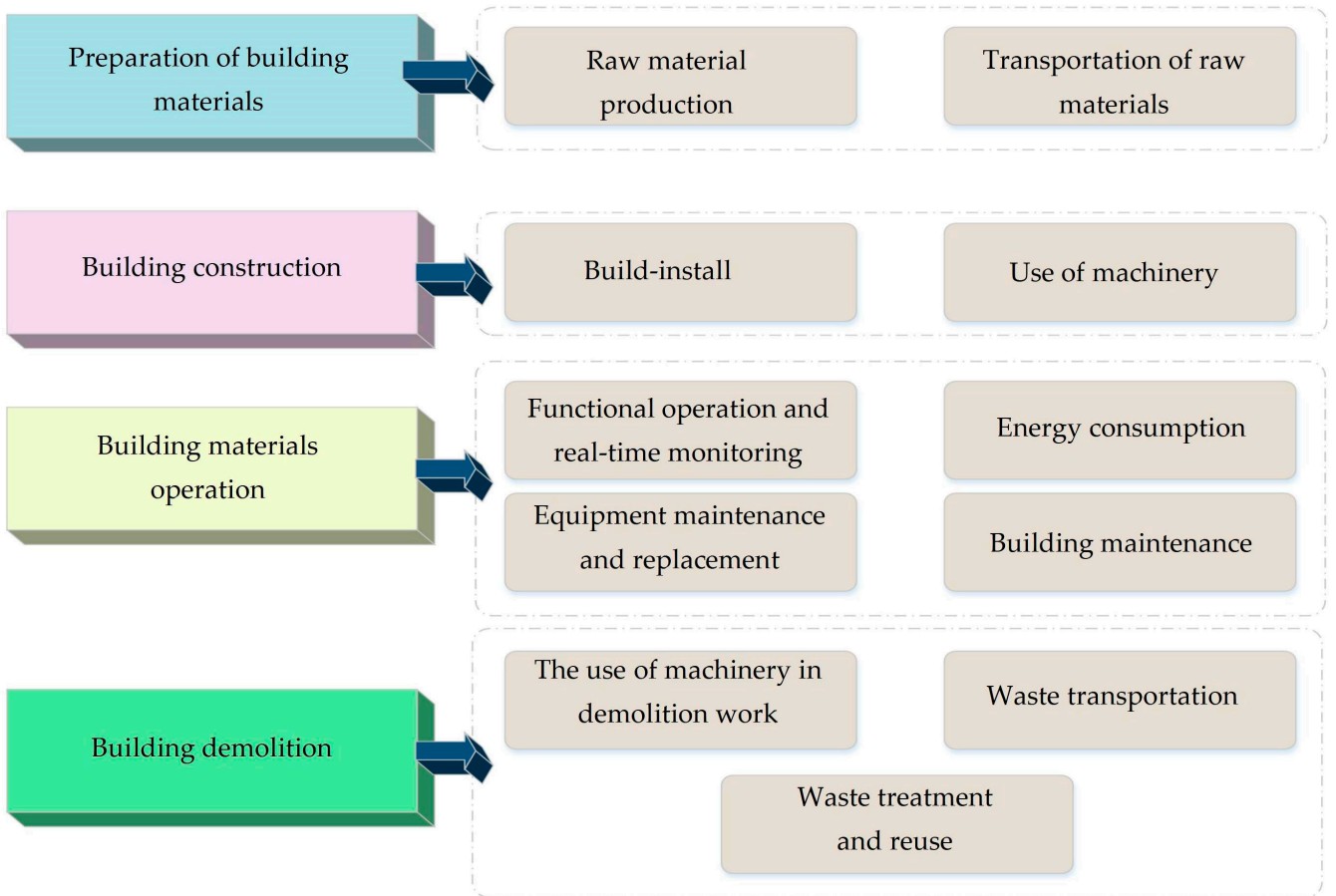

**Figure 1.** Design energy efficiency index of carbon neutrality in a building's life cycle.

**Table 1.** Design path of building carbon neutrality.

| Building Process | Implementation Path |
| --- | --- |
| Production stages | Decarburization of traditional building materials, application of low-carbon building materials, decarburization of building materials transportation |
| Construction stage | Intelligent construction, prefabrication technology, four-stage one environmental protection |
| Operation stage | Low-carbon building design, renewable energy utilization, and building energy-saving transformation |
| Demolition stage | Optimize the scheme of building recycling and demolition |

*2.2. Function System of Energy Efficiency for Carbon Neutrality Design of Whole Life Cycle Buildings*

Carbon-neutral design refers to a systematic method that makes the level of carbon-neutral design of buildings technologically advanced, environmentally friendly, and economically reasonable with the help of all kinds of big data related to it in the life cycle of buildings and various innovative design theories such as concurrent design. In order to

quantitatively express the basic level, development potential, and comprehensive effectiveness of the carbon-neutral design and comprehensively reflect and evaluate the true level of carbon-neutral energy efficiency in building a full-cycle carbon-neutral design, an "energy efficiency function" is proposed and constructed. The purpose is to characterize the carbon-neutral energy efficiency level, carbon-neutral potential, and operating mechanism of carbon-neutral strategy in building full-cycle design from a macro perspective. Figure 2 shows the three-dimensional model of the energy efficiency function of the carbon-neutral design.

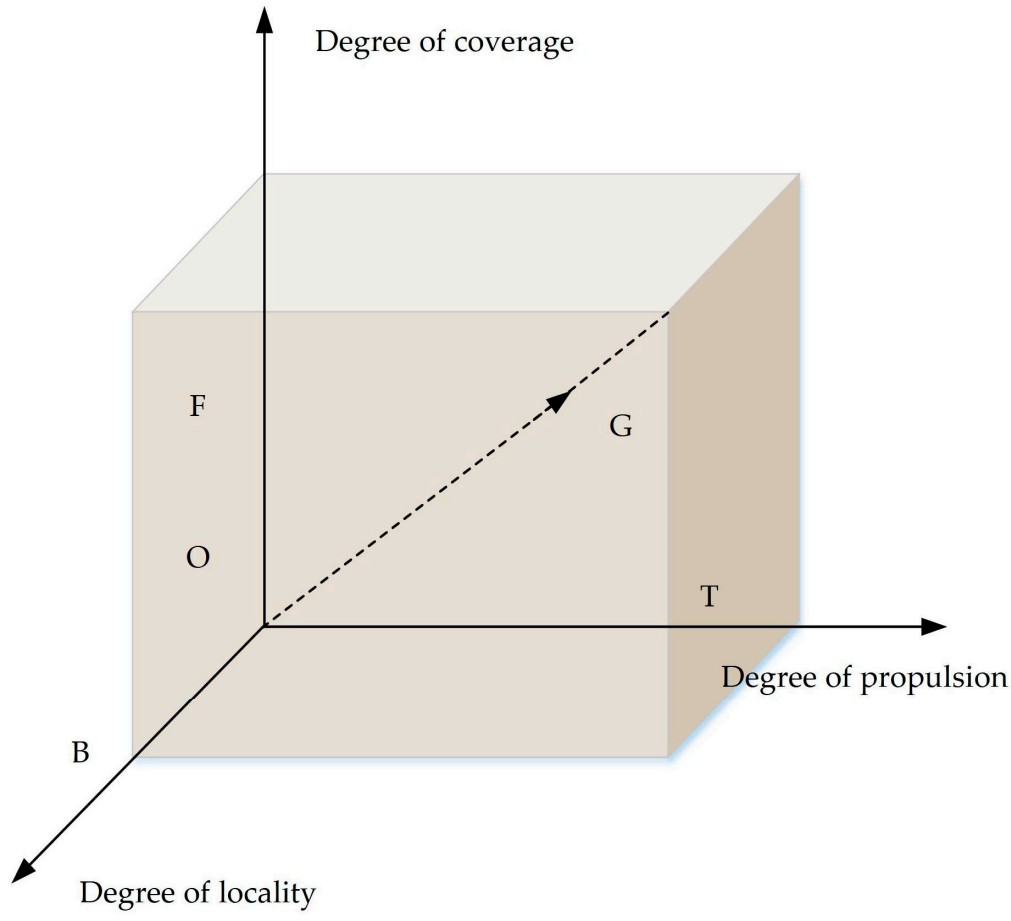

**Figure 2.** Three-dimensional model of energy efficiency function for the carbon-neutral design.

Figure 2 establishes a three-dimensional model of the energy efficiency function of the carbon-neutral design in a three-dimensional Cartesian coordinate system. The vector from *O* to *G* represents the best strategy of carbon-neutral design for buildings in the sense of specification, that is, the energy efficiency of carbon-neutral design. Any deviation or deviation from this vector is considered to be a mistake in carbon-neutral design strategy to varying degrees. In this paper, the design dimensions of carbon neutrality are divided into background, propulsion, and coverage. The background of the carbon-neutral design indicates that the building has the basic ability of carbon-neutral in the whole cycle, which mainly depends on two aspects: the quantity and quality of carbon-neutral design technical elements and carbon-neutral policy and capital investment. Based on the concept of "production function" in economics, this paper constructs and puts forward the background function of carbon-neutral design and makes an integrated analysis of the technical elements, policies, and capital investment of carbon-neutral design. In the Cobb–Douglas production function in the field of economics, the main factors that determine the development level of the industrial system are labor input, fixed assets, and comprehensive technical level. Cobb–Douglas production function will produce different

types of production functions according to the sum of the elastic coefficients of labor output and capital-output, but it is required that they are nonlinear. Similar to the calculation of production function in economics, this paper introduces the concept of input-output in carbon neutrality; that is, the background output is calculated through the input of carbon neutrality design technology and policy and capital. The mathematical expression of the proposed background function is shown in Equation (1):

$$R = G_t \times L^{\alpha} \times S^{\beta} \times m \tag{1}$$

In Equation (1), $R$ represents the carbon-neutral design background. $G_t$ is carbon-neutral quality index. $L$ is the proportion of engineering and technology. $S$ is the proportion of investment in research and development (R&D) of policies and funds. $\alpha$ is the input-output elastic coefficient of engineering standard. $\beta$ is the elastic coefficient of R&D capital input and output. $m$ is a random factor.

The promotion degree of the carbon-neutral design reflects the promotion degree of the overall design to enhance the carbon-neutral design of buildings in the whole cycle. This paper uses the concept of "elastic coefficient" for reference to construct and design the promotion degree function of carbon-neutral design. The elastic coefficient represents the ratio of the growth rates of two interrelated economic indicators in a certain period, and it is a measure of the dependence of the growth rate of one variable on the growth rate of another variable. The dependence between the two variables reflected by the elastic coefficient is reflected in the relationship between energy consumption and carbon emissions. According to the above conceptual method, it is extended to the design of the expression of carbon neutrality propulsion function, and the dependence between carbon emissions and energy consumption is calculated from the spatial dimension and the time dimension, respectively. According to this, the design propulsion function of carbon neutrality in the spatial dimension is shown in Equation (2):

$$P_s = e^{-\mu(\frac{C_i}{C} + \frac{E_i}{E})} \tag{2}$$

In Equation (2), $P_s$ reflects the matching relationship between the proportion of carbon emissions and the proportion of energy consumption in a building area. $C_i$ and $C$ represent the carbon emissions of region i and the total carbon emissions of all regions, respectively, and $E_i$ and $E$ represent the energy consumption of region i and the total energy consumption of all regions. $\mu$ is the correction coefficient, indicating the proportion of the total GDP of region i and the total GDP of all regions. In the time dimension, the expression of carbon-neutral propulsion degree is shown in Equation (3):

$$P_t = \mu \Delta C_i / \Delta E_i \tag{3}$$

In Equation (3), $P_t$ reflects the dependence between the change rate of pollutant discharge and the change rate of energy consumption in a country or region. $\Delta C_i$ and $\Delta E_i$ represent the annual change rate of pollutant discharge and energy consumption in region i, respectively.

Carbon-neutral design coverage indicates the distribution breadth and depth of design intervention of carbon emissions at various stages of the building. The distribution depth emphasizes the completeness of carbon-neutral emissions in all stages of the building to meet the design standards of carbon neutrality. The distribution breadth not only pays attention to the distribution of carbon neutrality in all stages of the building but also emphasizes the energy efficiency of regional carbon neutrality design. Based on the concept of "niche width" in ecology, this paper puts forward and constructs the coverage function of carbon-neutral design. In the field of biology, "niche breadth" refers to the sum of various resources used by organisms, and it can usually be accurately described by multi-dimensional space. However, the application field of this theory has already gone beyond the ecological category and penetrated into the related fields of economy and design. In

this paper, this field is applied to the field of carbon-neutral design, and the calculation formula of niche width is improved in combination with the requirements of the coverage of carbon-neutral design on the distribution breadth and width of buildings at various stages and the coverage function of carbon-neutral design is shown in Equations (4)–(6):

$$N_{\mathrm{i}} = \int_{\mathrm{i}-1}^{\mathrm{i}} \frac{n}{c} dt \tag{4}$$

$$P_{\mathrm{i}} = \frac{N_{\mathrm{i}}}{N_1 + \cdots + N_{\mathrm{i}} + \cdots N_s} \tag{5}$$

$$E = \mu \times \frac{1}{\sum_{\mathrm{i}=1}^{s} (P_{\mathrm{i}})^2} \tag{6}$$

In the above equation, $E$ represents carbon-neutral design coverage. $P_{\mathrm{i}}$ is the proportion of carbon neutralization design in the whole life cycle of the first cycle of the building. $\mu$ is the ratio parameter of the number of carbon-neutral design buildings, that is, the ratio of the number of regional carbon-neutral design buildings to the total number of carbon-neutral design buildings. $N_{\mathrm{i}}$ is the design integrity of carbon neutrality in the first cycle of the building. $s$ is the total life cycle of the building. $c$ is carbon-neutral design standard quantity. $n$ is the implementation amount of carbon-neutral design.

According to the three-dimensional theory of the carbon-neutral design energy efficiency function, the carbon-neutral design energy efficiency function is affected by the background, propulsion, and coverage functions of the carbon-neutral design at the same time, and the specific theoretical analysis is shown in Equation (7):

$$OG = \sqrt{\frac{OB^2 + OT^2 + OF^2}{3}} \tag{7}$$

In Equation (7), $OB$ represents the "background" of carbon-neutral design in the whole cycle of buildings, which mainly applies the Cobb–Douglas variant function in the field of carbon-neutral design to integrate the whole cycle cost index of buildings, engineering standards, and R&D investment ratio, with a view to representing the background of carbon-neutral design in the whole cycle of buildings. $OT$ represents the "propulsion degree" of carbon-neutral design in the whole cycle of buildings, which mainly uses the calculation method of "elasticity coefficient" in economics to calculate the elastic relationship between carbon emissions and energy consumption in the whole cycle of buildings from two dimensions of space and time, so as to realize the representation of the propulsion of carbon-neutral design in the whole cycle of buildings. $OF$ represents the "coverage" of carbon-neutral design in the whole cycle of buildings, mainly using the idea of "niche" in ecology to describe the advanced nature and quantity of carbon-neutral technology adopted in the whole cycle of buildings, hoping to realize the representation of the distribution breadth and width of carbon-neutral design in the whole cycle of buildings. $OG$ represents the comprehensive energy efficiency of building full-cycle carbon-neutral design, which not only considers the development basis of building full-cycle carbon-neutral design but also pays attention to the promotion level and coordination of building full-cycle carbon-neutral design in parallel, that is, "energy efficiency of carbon-neutral design" is the path optimization representation of the background, promotion, and coverage functions of carbon-neutral design in a balanced state.

Figure 3 illustrates the process for calculating building energy efficiency and carbon emissions.

Top-down statistical methods analyze the macroeconomic relationship between energy and the economy from the inside out, as shown in Figure 3, and are frequently based on historical time series data on national energy use and carbon emissions. Only after evaluating the building's overall energy consumption and carbon emissions is the time

and space downscale study carried out. The bottom-up statistical method is used to first calculate the hourly energy consumption and carbon emissions of a single building, and then the data is aggregated at the regional level. Forecasts and simulations of the building energy demand at regional, regional, and even national scales are made based on the typical building energy consumption as well as details like temperature and humidity, building performance, terminal equipment and operation characteristics, and building performance. The advantages and disadvantages of several methods for calculating the energy efficiency and carbon emissions of buildings are shown in Table 2.

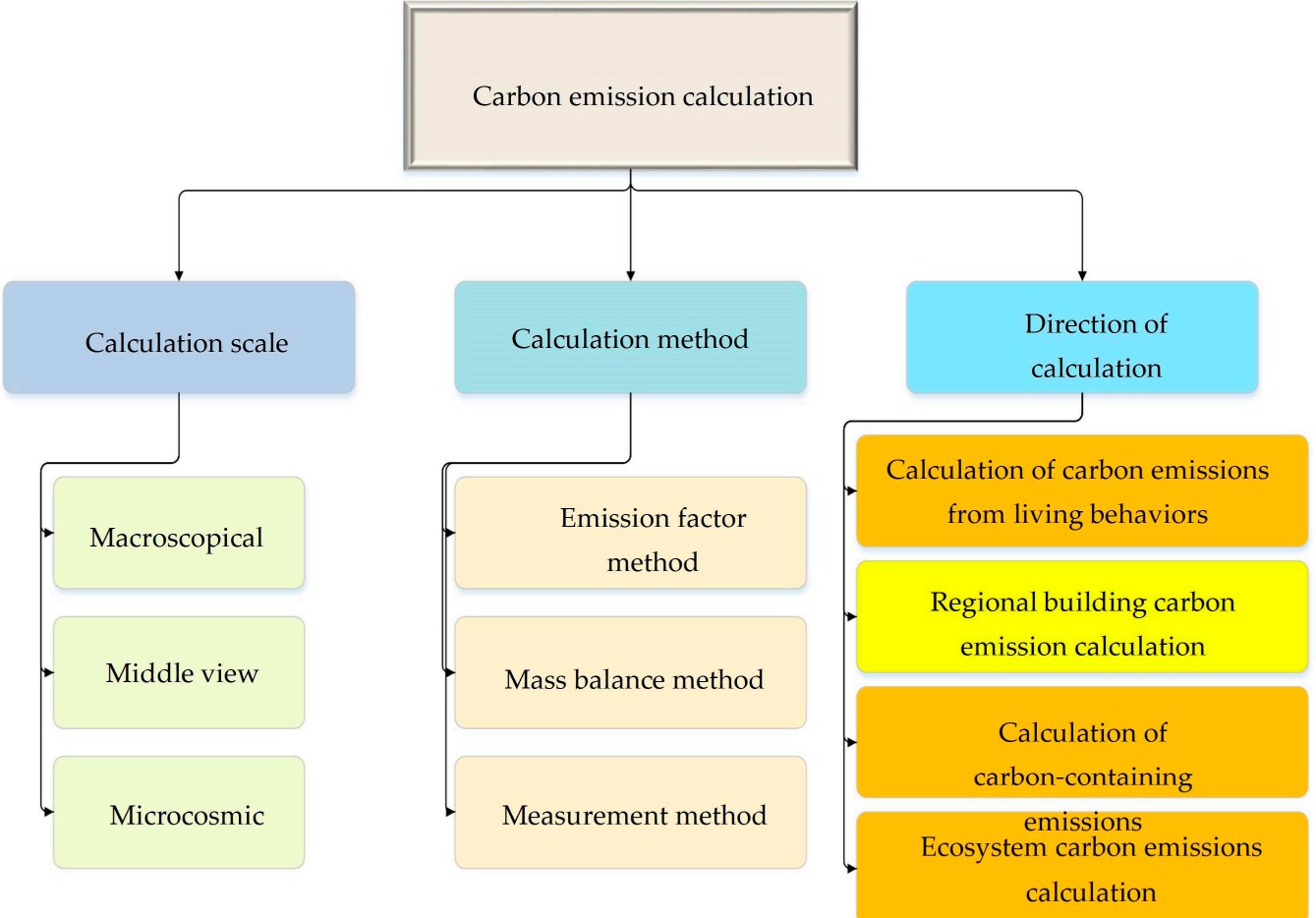

**Figure 3.** Calculation method of building energy efficiency and carbon emission.

**Table 2.** Advantages and disadvantages of calculation methods for building energy efficiency and carbon emission.

| Calculation Method | Advantage | Disadvantage |
|---|---|---|
| Top-down model [21] | (1) Pay attention to how the economy and energy interact<br>(2) The impacts of energy and emission policies and scenarios with different social costs and benefits can be simulated and analyzed.<br>(3) Based on macroeconomic data | (1) Forecast the future trend according to the interaction of the energy economy in the past.<br>(2) Lack of technical details<br>(3) It is not applicable to the analysis of specific technical policies.<br>(4) It is assumed that the market is efficient and inefficient. |

**Table 2.** *Cont.*

| Calculation Method | Advantage | Disadvantage |
|---|---|---|
| Bottom-up statistical model [22] | (1) Consider the macroeconomic and socio-economic impacts. (2) The typical terminal energy consumption can be determined. (3) Easy to develop and use (4) No detailed data is needed. | (1) Unable to provide a large number of data, poor flexibility. (2) The ability to analyze the effect of energy-saving measures is limited. (3) Dependence on historical energy consumption data (4) Need a large sample (5) Multicollinearity |
| Bottom-up physical model [23] | (1) Detailed description of current and future technologies (2) Use measurable physical data (3) Improve the effectiveness of energy consumption policy making. (4) Evaluate and quantify the impact of different technology combinations on energy supply and consumption. (5) Under the given demand, estimate the combination of technical measures with the lowest cost. | (1) Lack of description of market activities (2) Ignoring the relationship between energy consumption and macroeconomic activities (3) Need a lot of technical data. (4) Human behavior patterns cannot be defined in the model. |

In Table 2, the top-down method can focus on the overall scale of energy consumption, but it cannot reflect the technical details. The statistical analysis method is fast and simple, but it cannot reflect the characteristics of the building system and lacks explanatory power. Typical building methods can accurately describe the building system, but the modeling is complex and time-consuming.

### 2.3. Carbon-Neutral Energy Efficiency Model of Whole Life Cycle Building

The Intergovernmental Panel on Climate Change (IPCC) provided carbon accounting factors in 2006, and this paper selects appropriate carbon emission variables to develop a carbon balance accounting list based on those factors [24]. A value of 2.660 (kgCO$_2$/set) is the coal emission factor for construction traffic resources. The carbon emission factor from petroleum is 2.136 (kgCO$_2$/unit). Natural gas has a carbon emission factor of 1.657 (kgCO$_2$/unit). The carbon emission factor for power plants is 0.884 (kgCO$_2$/set). Gasoline has a carbon emission factor of 2.031 (kgCO$_2$/set). Kerosene has a carbon emission factor of 2.095 (kgCO$_2$/unit). Diesel oil has a carbon emission factor of 2.171 (kgCO$_2$/set). The carbon dioxide emission factor for gas is 1.301 (kgCO$_2$/unit). The building's entire life-cycle carbon neutrality and energy efficiency model is depicted in Figure 4.

The life cycle building carbon neutrality energy efficiency model shown in Figure 4 attempts to reduce energy use and carbon emissions. From a carbon neutrality standpoint, parameterized life cycles and carbon-integrated buildings are used to drive the best building design by integrating the carbon neutrality target into the carbon neutrality performance of buildings. The information correlation between architectural design parameters and carbon neutrality goals is investigated on the basis of identifying the numerical limits of architectural design parameters. Using hidden factors (K value, air tightness), target control indicators (figure coefficient, carbon fixation ratio, and Ouyang coefficient), and man–machine interaction technology, the man–machine cooperation between the computer and the architect in the construction process is realized, leading to the generation of a carbon neutrality energy-saving scheme.

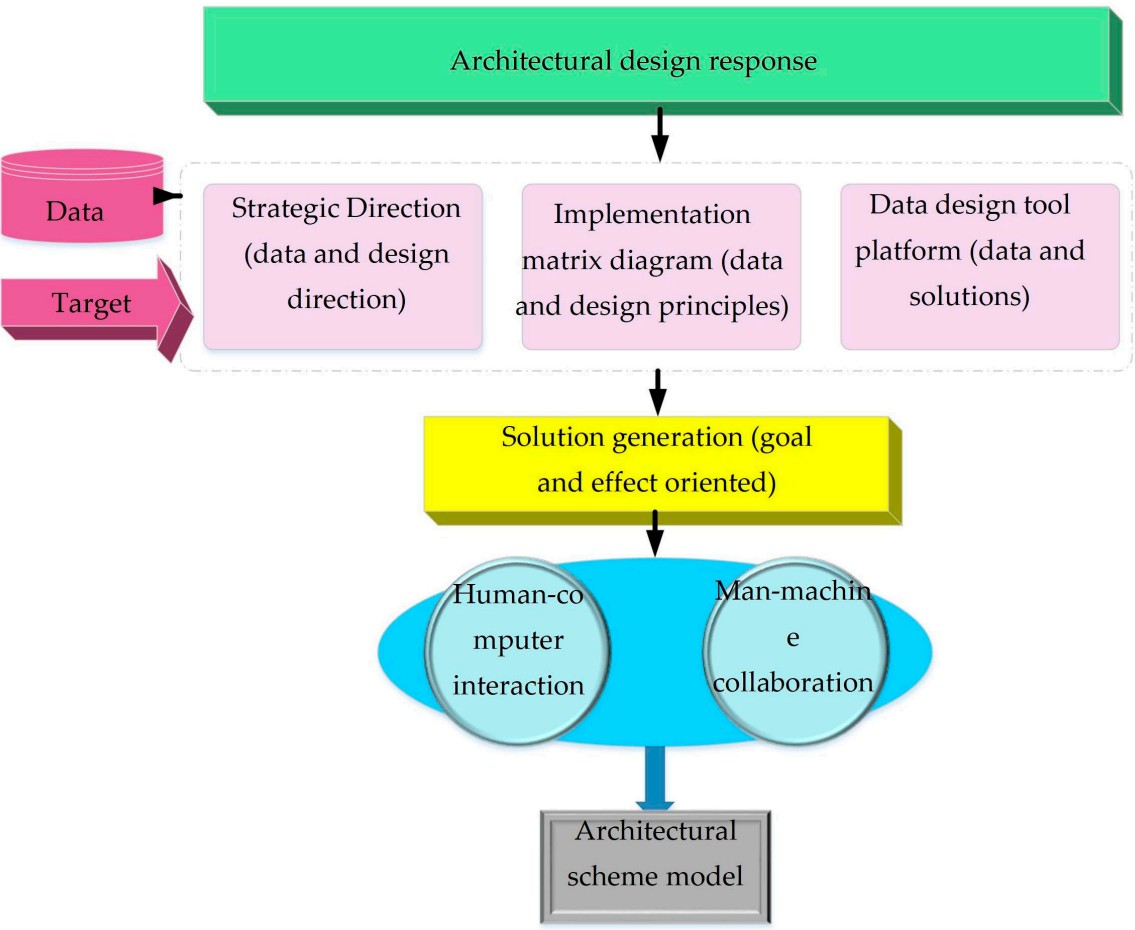

**Figure 4.** Carbon-neutral energy efficiency model of the whole life cycle of a building.

It is possible to calculate the greenhouse gas emissions from a building's entire production process using the carbon output of a single structure. This paper breaks down the building life cycle into four stages: manufacturing of building materials, building construction, building usage and maintenance, and building demolition, and then calculates the carbon emissions associated with each stage [25,26]. The calculation equation for carbon accounting across a building's whole life cycle is shown in Equation (8):

$$C = C_1 + C_2 + C_3 + C_4 \tag{8}$$

In Equation (8), $C$ represents the total carbon emission accounting amount (kg) of the building's carbon neutrality over the course of its whole life cycle. The amount of carbon emissions that are accounted for as being produced during the production of construction materials is $C_1$ (kg). $C_2$ represents the quantity of carbon emissions that were accounted for during the construction phase (kg). The amount of carbon emissions that are accounted for as being produced by buildings during use and maintenance is $C_3$ (kg). The accounting amount of carbon emissions (kg) from building demolition's waste treatment phase is $C_4$. Carbon emission index parameter setting in building materials production process, steel carbon emission: 1.7 kg; Carbon emission of steel bar: 2.0 kg; Carbon emission from cement: 0.9 kg; Carbon emission from glass: 23.45 m$^2$; Carbon emission of concrete: 400.67 m$^3$; Carbon emission of plastering mortar: 378.42 m$^3$; Carbon emission of aerated concrete block: 250.0 m$^3$; Thermal insulation carbon emission of polystyrene particles: 17.29 kg; Carbon emission of bamboo plywood: 33.10 m$^2$; Carbon emission of building tiles: 16.90 m$^2$; Aluminum and carbon emissions: 9.5 kg; Lime carbon emission: 1.2 kg; Standard brick carbon emissions of 504.0 thousand; Carbon emission of external wall elastic

coating: 2.6 kg; Asphalt treated using styrene-butadiene-styrene (SBS) emits 12.95 m$^2$ carbon dioxide. Equation (9) illustrates the simple equation for carbon emissions during the construction materials production process.

$$C_1 = \sum_{i=1}^{n} e_i \times q_i \tag{9}$$

In Equation (9), i stands for building energy, $e_i$ stands for the carbon emission of the i-type energy in the production stage of building materials, and $q_i$ stands for the i-type energy consumption in the production stage of building materials. Equation (10) shows the calculation expression of carbon emissions generated in the construction stage:

$$C_2 = C_{2,1} + C_{2,2} + C_{2,3} \tag{10}$$

The carbon emission produced during the manufacture of construction materials is represented by the symbol $C_{2,1}$ in Equation (10). The carbon emissions generated during the transportation of building resources are represented by $C_{2,2}$ while the emissions produced during the construction phase are represented by $C_{2,3}$. The calculation expression for carbon emissions produced during the manufacture of construction materials is shown in Equation (11):

$$C_{2,1} = \sum_{i=1}^{n} p_i \times q_i \times (1 - a_i) \tag{11}$$

In Equation (11), the terms i and "$p_i$" stand for the kind of building material and the emission index of Class I building material, respectively. The consumption of class i building materials is represented by $q_i$. The recovery rate of class i building materials is represented by the number $a_i$. Steel has a recycling coefficient of 0.8, aluminum has a recovery coefficient of 0.85, copper has a recovery coefficient of 0.90, and rebar has a recovery coefficient of 0.40. The estimate of carbon emissions produced during the transportation of building materials is shown in Equation (12):

$$C_{2,2} = \sum_{i=1}^{n} q_i \times s_i \times p_i \times k \tag{12}$$

The fuel consumption of Class i vehicles used for moving building resources is represented by $q_i$ in Equation (12). The kilometers traveled by Class i vehicles when transporting building materials are represented by the symbol $s_i$. In the transportation of building materials, the carbon emission and energy consumption of Class I modes of transportation are represented by the integer $p_i$. The kilogram to thyme conversion factor is represented by $k$. The computation of carbon emissions produced during the manufacturing stage of building materials is shown in Equation (13):

$$C_{2,3} = \sum_{i=1}^{n} q_i \times r_i \times p_i \tag{13}$$

$q_i$ in Equation (13) represents the energy consumption of Class i construction machinery in the construction stage. $r_i$ represents the number of Class i construction machinery in the construction stage. $p_i$ represents the carbon emission of Class i construction machinery in the construction stage. Set the diesel carbon emission index: 104.078 (kgCO$_2$/set); Carbon emission index of crane: 70.015 (kgCO$_2$/set). Equation (14) shows the calculation of carbon emissions generated during the use and maintenance of buildings:

$$C_3 = C_{3,1} + C_{3,2} \tag{14}$$

$C_{3,1}$ in Equation (14) stands for the amount of energy used to operate and maintain the structure. The carbon output from renovations during the building's usage and maintenance phase is represented by the number $C_{3,2}$. The computation of energy consumption during building use and maintenance is shown in Equation (15):

$$C_{3,1} = \sum_{i=1}^{n} q_i \times m \times p_i \tag{15}$$

In Equation (15), variable $q_i$ stands for the annual average Class i energy use during building use and maintenance. The building and maintenance equipment's service life is indicated by $m$. The carbon emission index of class i energy utilized in the building usage and maintenance stage is represented by the integer $p_i$. The greenhouse gas emissions created by numerous maintenance and transformations during the usage period are referred to as the "carbon emission of renovation in the use and maintenance stage of buildings". The types and quantities of building materials consumed by the maintenance and renovation project during the use period and during the use of mechanical shifts based on existing building maintenance and renovation records are calculated. The following equation calculates the carbon emissions generated during the waste treatment stage of building demolition:

$$C_4 = C_{4,1} + C_{4,2} + C_{4,3} \tag{16}$$

The energy use and carbon emissions produced by machines during the building demolition stage are represented by the variables $C_{4,1}$ in Equation (16). $C_{4,2}$ is a representation of the energy use and carbon emissions produced during the transportation of waste at the stage of building destruction. The carbon emissions caused by recyclable materials during the building demolition process are represented by the number $C_{4,3}$.

## 3. Results and Discussion

### 3.1. Analysis of Carbon Emissions in Different Stages of Building Carbon Neutralization

Figure 5 depicts the carbon emissions at various stages of building carbon neutralization. It indicates that 11,506.3 t, or 7.83% of the total output throughout the course of the full life cycle, of carbon dioxide was released during the construction period. During the duration of a building's entire life cycle, use and maintenance produce 134,492.1 t of carbon emissions or 91.47% of all emissions. Carbon emissions from building demolition and recycling processes total 1029.8 t, or 0.70 percent of all emissions during the course of a building's life cycle. Finally, it can be determined that the carbon emission per unit area of the building is 0.0583 t/year/m², which can be utilized as a reference data base for the building's carbon neutrality. The total carbon emission over the course of the building's life cycle is $C = 147,028.2$ t. Additionally, there is a significant variation between building carbon and carbon emission at every step of the building life cycle, which may be attributed to the influencing factors at each stage.

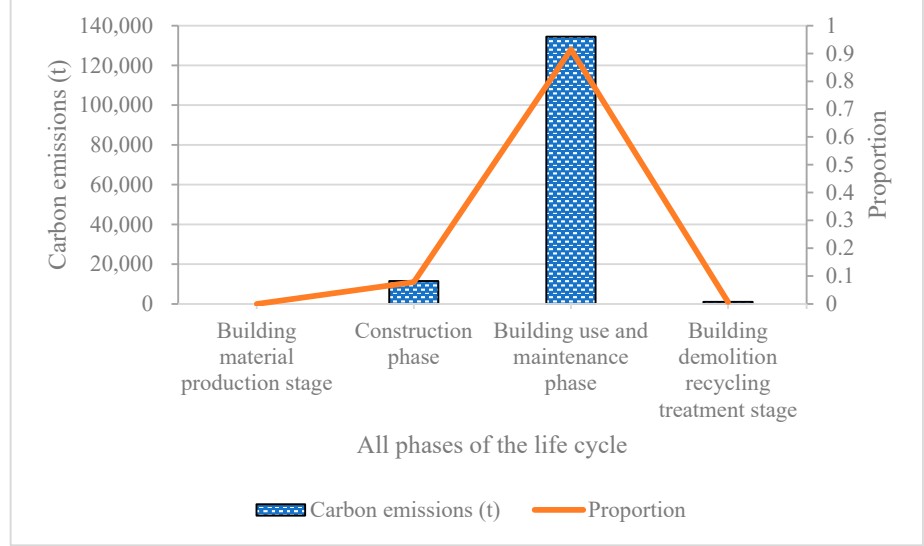

**Figure 5.** Carbon emissions at various stages of building carbon neutralization.

The computation of building carbon and carbon emissions throughout the course of the building's whole life cycle stage results in Figure 6, which shows the carbon emissions per

unit area for the building's complete life cycle stage. According to the calculation of carbon emissions per unit area in the entire life cycle phase of 10 buildings, the average carbon emissions per unit area in the production stage of building materials are $899.32 \, \text{kg/m}^2$ in the construction stage, $16{,}807.78 \, \text{kg/m}^2$ in the construction use and maintenance stage, $198.56 \, \text{kg/m}^2$ in the construction demolition stage, and $-4793.43 \, \text{kg/m}^2$ in the construction recovery stage.

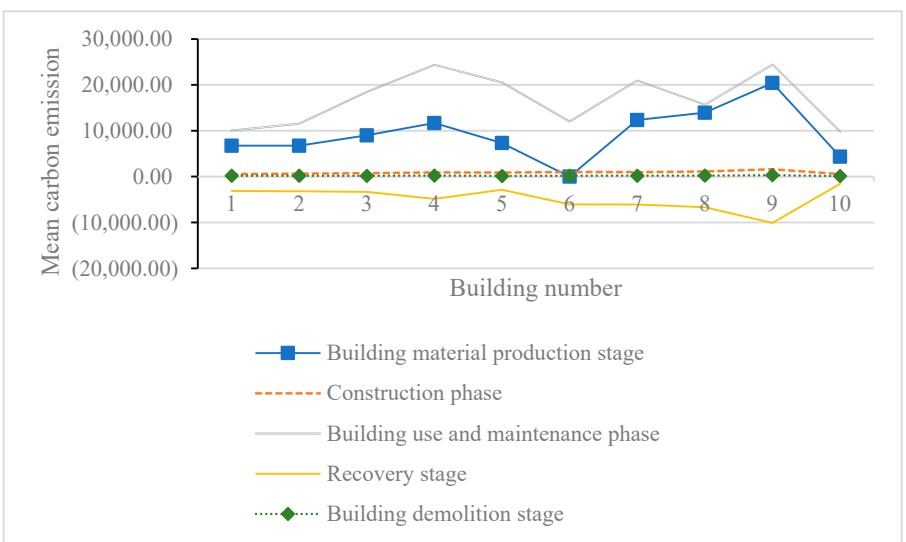

**Figure 6.** Carbon emission per unit area in the whole life cycle of buildings.

*3.2. Effectiveness Analysis of Energy Efficiency Model for Carbon-Neutral Design of Buildings*

The 22-story building's energy efficiency is examined in this section as it is being built. Figure 7 displays the energy effectiveness throughout construction. It displays an examination of the building's energy efficiency and carbon neutrality for each floor. The technical energy efficiency is much better than the scale energy efficiency and comprehensive energy efficiency, and all efficiency values in the construction stage are less than or equal to 1.00 ("1.00" is the relative highest value, so "1.00" means strong). As a result, when decision-making units compare data, the higher the energy efficiency value during building, the lower the consumption, the higher the production, and the 100% capacity utilization rate. The least amount of technical energy is used during building, and nearly all of the available capacity is being used.

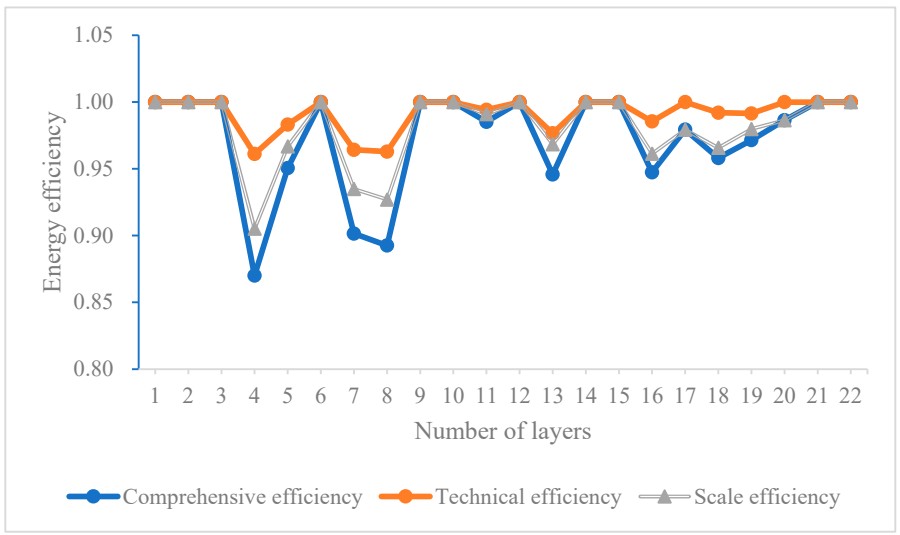

**Figure 7.** Energy efficiency in building construction stage.

Through the Data envelopment analysis (DEA) validity analysis of the energy efficiency model of building carbon-neutral design, it is indicated that "0" is invalid and "1" is strongly effective. Figure 8 shows the effectiveness of the energy efficiency model of building the carbon-neutral design. In Figure 8, the average comprehensive effectiveness of the energy efficiency model of building the carbon-neutral design is 0.98, and the energy efficiency model of building the carbon-neutral design is effective.

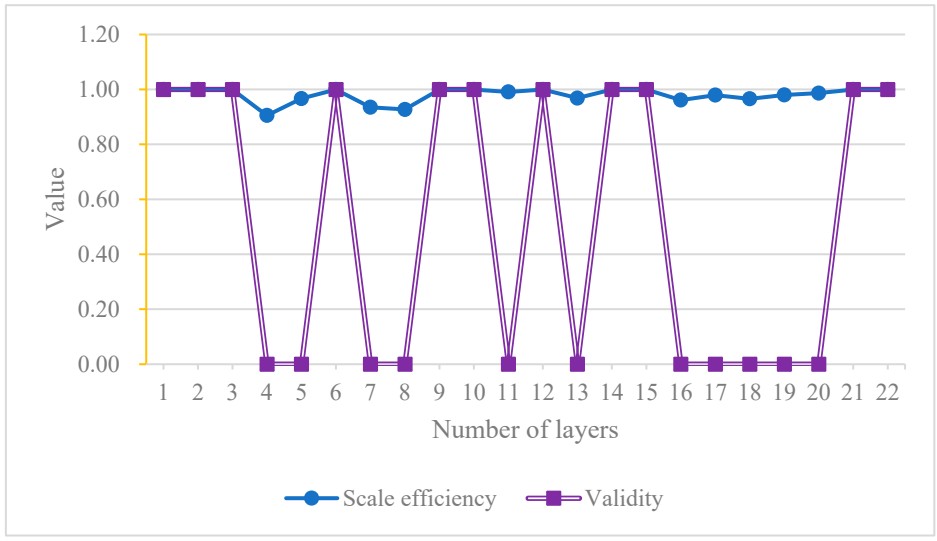

**Figure 8.** Validity of energy efficiency model for the carbon-neutral design of an average of 10 buildings.

## 4. Conclusions

One of the fundamental quantitative pillars of the novel system of developing carbon neutrality design concepts, theories, methodologies, and tools in this paper is the mathematical model of "energy efficiency". Combined with the actual situation of the development of national carbon-neutral building goals, the research on technical support measures for architectural design is completed. Based on this, an energy efficiency model can be implemented to quantitatively predict the carbon-neutral design scheme of buildings. By combining architecture, engineering, mathematics, and other disciplines, using the functional model of carbon neutrality design foundation, promotion, and coverage, and combining with the mechanism of carbon neutrality design concepts, the "function" model quantitatively studies the operating rules and principles of building carbon neutrality design efficiency. This is done by taking the theoretical modeling of building carbon neutrality design evaluation as the research object. The findings indicate that: (1) Technology energy efficiency is far better than scale energy efficiency and comprehensive energy efficiency, the way used, and technology spillover benefits are potential reasons for this change. Building carbon and carbon emission differ significantly across the whole life cycle stage, which can be attributed to the influencing elements in each stage of the building life cycle. Carbon neutrality design energy efficiency evaluation has a strong element impact effect at the stage, and carbon neutrality design has more balanced overall benefits at the stage of higher building technology capability. (2) When decision-making units compare data, the better an energy efficiency value inside the building stage, the less consumption and the higher the production, and the capacity utilization rate is 100%. Construction is when technical energy is used the least, and capacity utilization is extremely close to 100%. (3) With an average overall effectiveness of 0.98, the energy efficiency model of a building with a carbon neutrality design is highly effective. The data and calculation model presented in this research provides support for constructing an energy efficiency model for a carbon-neutral building design. One of the remaining flaws in the study methodology is that it is impossible to identify this relationship, even though the relationship between building carbon

and energy efficiency in all stages of the life cycle is studied, and the research on how the energy efficiency function is combined with the design strategy needs to be expanded, so as to further characterize the energy efficiency level, development potential and selection strategy of carbon neutrality design in various regions. Future studies will therefore need to focus on energy efficiency over the entire life cycle.

**Author Contributions:** Conceptualization, R.L. and L.H.; Methodology, R.L. and X.Z.; Software, X.Z. and J.L.; Validation, R.L., and X.Z.; Formal Analysis, X.Z. and J.L.; Investigation, X.Z. and J.L.; Resources, R.L.; Data Curation, X.Z. and J.L.; Writing—Original Draft Preparation, R.L., X.Z. and L.H.; Writing—review & editing, R.L., X.Z., L.H. and J.L.; Visualization, X.Z. and J.L.; Supervision, R.L.; Project Administration, R.L. and L.H.; Funding Acquisition, R.L. and L.H. All authors have read and agreed to the published version of the manuscript.

**Funding:** This article is a joint project between General projects of social science planning in Guangdong Province (GD20CYS15) and the 14th Five Year Plan for the Development of Philosophy and Social Science in 2021 in Guangzhou, with the approval number of No.: 2021GZGJ283.

**Institutional Review Board Statement:** Not applicable.

**Informed Consent Statement:** Not applicable.

**Data Availability Statement:** Data and materials are available from the authors upon request.

**Conflicts of Interest:** The authors declare no conflict of interest.

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
