# Peer review of "Energy Efficiency Model Construction of Building Carbon Neutrality Design"

_sustainability, doi:10.3390/su15129265_

Round 1
Reviewer 1 Report
This manuscript puts forward the energy efficiency function model of carbon neutrality design applied to the whole life cycle of buildings, systematically and comprehensively quantifies a feasible method for calculating carbon neutrality building energy efficiency, which is meaningful for improving the carbon neutrality design system of buildings and can reach the level of publishing journals.
However, if the following problems can be solved, the integrity and extension of the manuscript can be improved. Some comments below should be considered:
1. In the introduction, some papers about buildings should be referred:
https://doi.org/10.3390/ma16062137
https://doi.org/10.1108/09699980610659607
2. Figure 6, the legend part is too big. This figure should be improved.
3. Figure 7, From a strict standpoint, “1” should be changed to “1.00” and “0.8” should be changed to “0.80”. Similar problems also appear in other figures.
4. There are some equations and many symbols used, which should be carefully checked.
5. In the conclusion, it is recommended that the author further elaborate on the shortcomings of the research, in order to expand more possibilities and directions for future research.
6. 'IPCC' first appears in the abstract, it is recommended not to use abbreviations.
7. The abbreviation and type setting of the journal in references should be checked and corrected according to the demand of Sustainability. i.e. ref.[6], ref.[7], ref.[15], etc.
Moderate editing of English language.
Reviewer 2 Report
Reviewer’s comments of manuscript, sustainability-2422870, “Energy Efficiency Model Construction of Building Carbon Neutrality Design”
This work proposes a novel and effective strategy for energy efficiency model construction of building carbon neutrality design. More significantly, to develop a workable way of calculating energy efficiency in buildings that achieve carbon neutrality and the system for such a building's design energy efficiency function. The paper is of general interest for researchers working on the carbon neutrality design. The paper is consistently written and well organized. I recommend its publication after addressing following minor issues:
1. As the authors mentioned in the manuscript, building carbon and carbon emission differ significantly across the whole life cycle stage, why it can be attributed to the influencing elements in each stage of the building life cycle, and construction is when technical energy is used the least, and with an average overall effectiveness and a carbon neutrality design. Therefore, I strongly suggest the author should add the related statement into MS for further reading and understanding.
2. In addition to the workable way of calculating energy efficiency in buildings in this work, which model structure are expected to achieve carbon neutrality and the system for such a building's design energy efficiency function? Hereby, I strongly suggest the author should supplement the related statement to provide the directive evidence for the final explanation.
3. There are some minor errors need to be corrected. e.g. the italics of subscript “such as i” in all Eqs. In addition, the journal abbreviation and typesetting in references should be done strictly according to the demand and rule of Sustainability. i.e. ref.[6], ref.[7], ref.[15], ref.[22], ref.[25], etc.
4. English writing of this MS could be improved and polished further.
English writing of this MS could be improved and polished further.
Reviewer 3 Report
Presented paper “Energy Efficiency Model Construction of Building Carbon 2 Neutrality Design” expose a theoretical model for evaluating of carbon neutral design of a building. The authors well present concept of energy efficiency function of a building from row materials to completed project and then energy efficiency life style. The above specific terms are very well presented and detailed explained. It is well visible and understandable that the authors have performed thorough research. They have demonstrated professionalism in building and scientific skills.
Visualization is in good quality – all figures and diagrams are clear and understandable. All tables are clear and as per journal requirements.
All references are relevant to the topic of the manuscript.
Despite of the above I have the following remarks to the authors reviewing this article:
1. Structure of the Abstract is not correct. The authors need to expose briefly the problem and reason for writing this article.
2. It is not reasonable acronyms in the Abstract. The authors have to write full name. The acronyms have to be a part of the main body of the article.
Reviewer 4 Report
This manuscript offers an insightful and robust investigation into the carbon emissions at different stages of a building's lifecycle. The methodology is rigorous and the analysis is comprehensive, contributing to our understanding of carbon neutrality in the context of building design. The authors have clearly demonstrated a significant effort to quantify and analyze the carbon emissions and energy efficiency of buildings, which is critical for the development of sustainable urban environments. However, there are a few minor aspects that, if addressed, could improve the clarity and completeness of the manuscript. Suggest to be revised with minor revisions.
Technical Comments:
l The introduction part is suggested to be enriched with more general updated literature regarding Carbon neutrality from addition aspects, the article below is suggested to be consulted as starting point. Status and Outlook of Oil Field Chemistry-Assisted Analysis during the Energy Transition Period. Energy & Fuels 2022. Investigations of CO2 storage capacity and flow behavior in shale formation. Journal of Petroleum Science and Engineering 2022.
l Equation Clarification: For readers not familiar with the notations and symbols used in equations (15) and (16), it would be helpful to provide additional explanation for each variable. It would be beneficial to have a table or a separate section clearly defining all the variables.
l Graphical Representation: The descriptions of Figure 5 and Figure 6 were quite detailed, but without the actual figures, it's hard to fully understand the authors' explanations. It's recommended to include these figures or, if that is not possible, provide additional context or explanation in the text.
l Data Analysis: In the results and discussion section, specific data points and percentages are given, but it's not always clear how these were derived. Providing more detail about the data analysis methods would strengthen this section.
l Energy Efficiency Model Validation: The authors have done an excellent job discussing the energy efficiency model for carbon-neutral design of buildings. However, a more detailed explanation on the validation technique (Data envelopment analysis) and how the validity was determined as "0" for invalid and "1" for strong would be helpful for readers not familiar with this analysis.
l Building Carbon and Carbon Emission Relationship: The authors suggest in their conclusion that a significant variation exists between building carbon and carbon emissions at every step of the building lifecycle. This is an important finding, but more explanation or analysis is needed. How does this variation impact the overall carbon neutrality of a building? What could be the potential reasons for this variation?
l Future Work: In the conclusion, the authors briefly mention future work to focus on energy efficiency over the entire lifecycle. It would be helpful to provide more detail on this: what are the expected challenges? What methods or approaches will be used?
Addressing these points would add to the completeness and clarity of this already comprehensive study. The manuscript provides a valuable contribution to the field of carbon neutrality in building design, and these minor revisions could improve its overall quality.
Moderate editing of English language
Reviewer 5 Report
The review topic is interesting and the manuscript is well-written. Therefore, the manuscript has some problems that are listed below:
1) The abstract should be improved. Please, rewrite it. What is the novelty of your study? What is the impact of your work? The results are poorly shown in the abstract.
2) The paragraph (Lines 63-84) mentions many studies from you had some conclusions, however, no papers are mentioned in the text. The authors should rewrite it linking the literature about the topic.
3) Introduction. What is the novelty of your study? What is the aim?
4) In section 3, the authors should compare the results found with the ones in the literature. What is the impact of your study? How does it differ from the other design published previously?
Round 2
Reviewer 5 Report
The quality of the manuscript has increased significantly.